# Single-Atom Nanozymes: Fabrication, Characterization, Surface Modification and Applications of ROS Scavenging and Antibacterial

**DOI:** 10.3390/molecules27175426

**Published:** 2022-08-25

**Authors:** Haihan Song, Mengli Zhang, Weijun Tong

**Affiliations:** MOE Key Laboratory of Macromolecular Synthesis and Functionalization, Department of Polymer Science and Engineering, Zhejiang University, Hangzhou 310027, China

**Keywords:** nanozyme, single-atom nanozyme, surface modification, ROS scavenging, antibacterial

## Abstract

Nanozymes are nanomaterials with intrinsic natural enzyme-like catalytic properties. They have received extensive attention and have the potential to be an alternative to natural enzymes. Increasing the atom utilization rate of active centers in nanozymes has gradually become a concern of scientists. As the limit of designing nanozymes at the atomic level, single-atom nanozymes (SAzymes) have become the research frontier of the biomedical field recently because of their high atom utilization, well-defined active centers, and good natural enzyme mimicry. In this review, we first introduce the preparation of SAzymes through pyrolysis and defect engineering with regulated activity, then the characterization and surface modification methods of SAzymes are introduced. The possible influences of surface modification on the activity of SAzymes are discussed. Furthermore, we summarize the applications of SAzymes in the biomedical fields, especially in those of reactive oxygen species (ROS) scavenging and antibacterial. Finally, the challenges and opportunities of SAzymes are summarized and prospected.

## 1. Introduction

Enzymes play a very important role in thousands of biochemical reactions carried out in our body from moment to moment. Most enzymes are proteins, and a few are RNA [1], which can specifically catalyze the reaction under mild conditions, and have the characteristics of high catalytic efficiency and good specificity [2]. However, natural enzymes are easily inactivated under harsh conditions, together with complicated preparation process and high cost, which greatly limit their applications [3]. Therefore, people continually develop artificial enzymes as the substitutes of natural enzymes.

In 2007, Yan’s group [4] first discovered that Fe_3_O_4_ nanoparticles (NPs) had peroxidase (POD)-like activity, and their pH and temperature stability were much higher than those of natural horseradish peroxidase (HRP). Since then, the term “nanozyme” has entered the field of vision of scientists. Nanozymes are a class of nanomaterials with catalytic activity similar to natural enzymes, which have the advantages of high stability, low cost, and simple preparation [5]. Scientists have successfully developed a series of carbon-based nanozymes, metal-based nanozymes, metal oxide-based nanozymes, and so on [6], which are widely used in biosensing [7,8], antibacterial [9,10,11], anti-inflammatory [12,13,14,15], cancer treatment [16,17,18,19], and other biomedical fields.

Although nanozymes have a wide range of applications in the biomedical field, their catalytic activity, specificity, and affinity for substrates still need to be improved compared with natural enzymes [6,20]. The diversity of protein composition in natural enzymes and the complexity of the folding process endow them with fairly fine structures. Such kind of fine structures are normally absent in nanozymes; thus, their performance in catalytic reactions is far inferior to that of natural enzymes [21]. Furthermore, the internal crystal structure of nanozymes is not uniform and the elemental composition is complex, which makes it difficult to study their active sites, catalytic reaction unit, and catalytic mechanisms [22,23].

In 2011, Zhang’s group [24] proposed the concept of single-atom catalysts (SACs). They prepared a new type of catalyst with single Pt atoms atomically dispersed on the surface of iron oxide (FeOx) (denoted as Pt/FeOx). Pt/FeOx exhibited extremely high reactivity in CO oxidation and preferential oxidation. After years of development, SACs are defined as a novel catalyst in which catalytically active isolated metal atoms are immobilized on supports [25]. They combine the advantages of homogeneous catalysis and heterogeneous catalysis [26], and SACs possess the maximized atom utilization, unsaturated coordination of active sites, and adjustable electronic properties, which make them a hot research direction in the field of catalysis [27,28,29]. Meanwhile, the high atom utilization rate can also reduce the production cost of SACs [30] and the potential toxic and side effects of metal ions [31,32]. More importantly, the isolated and dispersed atoms in SACs are stabilized by surrounding ligands, which is very similar to the geometric structure in some natural enzymes centered on metal atoms [30]; thus, SACs also have been proposed for use as biomimetic single-atom nanozymes (SAzymes). In SAzymes there are well-defined and precisely controlled active sites, so it is easy to determine the catalytic reaction unit, which is particularly important for revealing the structure–activity relationship and studying the catalytic reaction mechanism [22,28].

In this review, we first systematically describe the preparation of SAzymes through pyrolysis and defect engineering and the regulation of their enzyme-like activities, followed by their characterization. In the following part, the strategies of surface modification for SAzymes are further introduced, which is critical for their biomedical applications. Then, the applications of SAzymes in the fields of reactive oxygen species (ROS) scavenging and antibacterial are discussed. Finally, the challenges and opportunities of SAzymes are summarized and prospected.

## 2. Preparation of SAzymes

Although SAzymes have broad applications in the biomedical fields, the preparation is not easy because of the high surface energy of isolated metal atoms which tend to aggregate into nanoclusters during the preparation process [28,33]. Therefore, preventing the aggregation of metal atoms during the preparation has become the top priority in the development of preparation strategies. Scientists have developed a variety of synthesis strategies, including pyrolysis, defect engineering, atomic layer deposition, photochemical reduction, and so on [28,30], and the advantages and disadvantages of these methods are listed in Table 1. In this review, we mainly focus on the widely used pyrolysis strategy and the defect engineering strategy based on wet chemistry.

### 2.1. Pyrolysis

As a widely used preparation method for carbon-supported SAzymes [37], pyrolysis is usually a method to prepare SAzymes by thermally pyrolyzing the carrier loaded with the active species precursor in a specified gas atmosphere, which can realize the precise control of structure and high loading of atomically dispersed metal atoms [30]. Classical pyrolysis usually includes three steps. Firstly, the active center precursor is loaded on/into the carrier, and then it is pyrolyzed in a gas atmosphere. During the cracking process, the carbon support is converted into N-doped carbon material and the active centers coordinate with N or C atoms to form an M–N/C structure (M represents a transition metal) [38], and finally the template or unreacted raw materials are removed.

For example, Zhu et al. [39] used aniline, ammonium persulfate, and SiO_2_ to synthesize a support, and then added the active center precursor Pd(CH_3_CN)_2_Cl_2_ on it. After pyrolysis in Ar, the template was washed away with NaOH, and Pd–C SAzyme with atomically dispersed Pd atoms as active centers was obtained. Pd–C SAzyme showed strong POD-like activity under acidic conditions and excellent photothermal properties. Cheng et al. [40] mixed oxidized carbon nanotubes with pyrrole, added ammonium persulfate to induce the polymerization of pyrrole into polypyrrole, and then added Fe(NO_3_)_3_ and NaCl which would promote the attachment of Fe species on the surface of carbon nanotubes. Afterwards, it was pyrolyzed at 900 °C in N_2_, and annealed in NH_3_ after removing NaCl and unreacted Fe species with H_2_SO_4_. Finally, the SAzyme with Fe as the active centers and carbon nanotubes as the carrier (denoted as CNT/FeNC) was obtained (Figure 1A). CNT/FeNC exhibited excellent POD-like activity, enabling sensitive detection of H_2_O_2_, glucose, and ascorbic acid (AsA).

In addition, metal–organic frameworks (MOFs) are also important support materials which are widely used in the preparation of SAzymes through pyrolysis. Composed of metal nodes and organic ligands, MOFs are a kind of porous crystalline material, and have high specific surface area, well-defined structures, and can be flexibly designed [27]. More importantly, the dispersed metal nodes in MOFs are well defined, which can be directly transformed into isolated metal sites on the carriers due to the carbonization of organic ligands during the pyrolysis process [20]. Therefore, MOFs are excellent carriers for the preparation of SAzymes [28]. The precursor containing the active center also can be easily immobilized in the MOF by ion exchange, etc., and then the metal atom of the active center is coordinated with N or C during the pyrolysis process and transformed into N-doped carbon materials (M–N/C) [30,41].

ZIF-8, with Zn as the metal node and 2-methylimidazole as the organic ligand, has a large specific surface area, numerous pores, and abundant N atoms, which is the preferred MOF for the preparation of SAzyme through pyrolysis [42]. Niu et al. [42] added Fe^3+^ during the synthesis process to obtain Fe-doped ZIF-8, which was subsequently pyrolyzed in N_2_ and NH_3_ at 900 °C to obtain the SAzyme with atomically distributed Fe as the active center and N-doped carbon material as the carrier (denoted as Fe–N–C SAN) (Figure 1B). Fe–N–C SAN showed good POD-like activity, its activity was comparable to that of HRP, and it was more stable, which could realize the highly sensitive detection of butyrylcholinesterase, a typical biomarker of organophosphorus pesticide exposure [43].

Porphyrin and its derivatives can generate cytotoxic singlet oxygen (^1^O_2_) under irradiation of appropriate light, and are commonly used as photosensitizers in cancer photodynamic therapy [44]. Liu’s group used mesoporous silica (m-SiO_2_) to coat ZIF-8 and then pyrolyzed to prepare monodisperse mesoporous carbon nanospheres containing porphyrin-like zinc centers (denoted as PMCS) (Figure 1C), in which Zn atoms were atomically dispersed. As a protective layer, m-SiO_2_, which could be removed by NaOH etching, will effectively prevent active centers of metal atoms from aggregating during the pyrolysis and ensure the successful synthesis of SAzymes [45]. The PMCS have strong absorption in the near-infrared region and can generate ^1^O_2_ under the irradiation of 808 nm laser. Using a similar strategy, Cao et al. [46] fabricated bimetallic MOFs by encapsulating Co species in ZIF-8, followed by pyrolysis in N_2_ to obtain PMCS with atomically dispersed Co as active centers (denoted as Co/PMCS) (Figure 1D). Co/PMCS could mimic superoxide dismutase (SOD), catalase (CAT), and glutathione peroxidase (GPx) to rapidly scavenge reactive oxygen and nitrogen species for the treatment of bacterial-induced sepsis.

**Figure 1 molecules-27-05426-f001:**
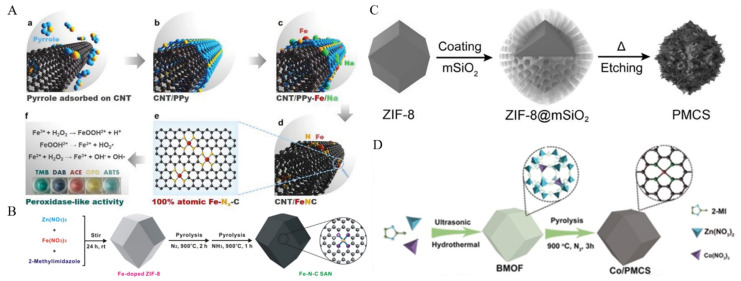
Preparation of SAzymes by pyrolysis. (**A**) CNT/FeNC. (a) Pyrrole adsorbed on CNT. (b) Polypyrrole coated on CNT to form CNT/PPy. (c) Metal cations adsorbed on CNT/PPy. (d) Pyrolysis to form CNT/FeNC. (e) Structure of Fe-Nx-C. (f) POD-like activity of CNT/FeNC. [40]. Copyright 2019, Wiley-VCH. (**B**) Fe–NC SAN [42]. Copyright 2019, Elsevier. (**C**) PMCS [45]. Copyright 2016, Wiley-VCH. (**D**) Co/PMCS [46]. Copyright 2020, Wiley-VCH.

The activity of SAzymes can be adjusted by control of the temperature and gas atmosphere in pyrolysis. Xu et al. [47] prepared PMCS with atomically dispersed Zn as active sites by pyrolysis of ZIF-8, and adjusted the pyrolysis temperature from 600 °C to 1000 °C to obtain a series of SAzymes (denoted as c-ZIF-600, c-ZIF-700, PMCS, c-ZIF-900, and c-ZIF-1000). It was found that c-ZIF-600 and c-ZIF-700 had almost no POD-like activity, c-ZIF-900 and c-ZIF-1000 had weak POD-like activity, and only PMCS exhibited higher POD-like activity. This is due to the increase of the structural defects in pyrolyzed ZIF-8 when the temperature rises. The defects can accelerate the diffusion of substrates to the active sites, thereby improving the catalytic activity. However, the increase of pyrolysis temperature also results in the decrease of Zn, which is the active center. On the whole, the POD-like activity of PMCS obtained by pyrolysis at 800 °C is the highest. The gas atmosphere of pyrolysis also can greatly influence the activity of SAzymes. For example, Wang et al. [48] used TiO_2_ as carrier to prepare Co/TiO_2_ SAzymes protected by SiO_2_ through pyrolysis. By changing the atmosphere of pyrolysis, they obtained two kinds of Co/TiO_2_ SAzymes denoted as Co/A–TiO_2_ (pyrolysis in air) and Co/N–TiO_2_ (pyrolysis in N_2_). The oxidase-like (OXD-like) activity of Co/A–TiO_2_ was significantly higher than that of Co/N–TiO_2_, and Co(II)/Co(III) in Co/A–TiO_2_ was higher than that of Co/N–TiO_2_, which was also the fundamental reason why the enzyme-like activity of Co/A–TiO_2_ was higher than that of Co/N–TiO_2_. The different pyrolysis atmospheres can change the coordination environment of the active center, thereby affecting the activity of SAzymes.

Cytocrome P450 and HRP are involved in a variety of biochemical reactions in the body, and their active center is a heme group containing Fe, in which Fe is coordinated by four N atoms in the plane, while it is also coordinated axially with a S or N atom [49]. Therefore, the scientists prepared FeN_5_ SAzymes by imitating the structure of cytocrome P450 and introducing axial N coordination into Fe SAzymes. Compared with the common Fe–N_4_ SAzymes, the enzyme-like activity of FeN_5_ SAzymes was greatly improved [50]. In this study, iron phthalocyanine (FePc) was encapsulated into Zn–MOF to form the host–guest structure of Fe@Zn–MOF, followed by pyrolysis in N_2_ to obtain a five-coordinated Fe SAzyme (denoted as FeN_5_ SA/CNF) (Figure 2). Its fine structure was characterized by X-ray absorption near-edge spectroscopy (XANES) and extended X-ray absorption fine structure (EXAFS) spectroscopy. The results confirmed the formation of the Fe–N_5_ structure. The OXD-like activity of FeN_5_ SA/CNF and the initial reaction rate were much higher than those of FeN_4_ SA/CNF and its counterparts with different metal ions, indicating that the axial N-coordination structure and the type of metal active center were equally important to its activity. Similarly, Xu et al. [51] prepared FeN_5_ SAzyme using a melamine-mediated two-step pyrolysis method, in which melamine played a role in providing the axial direction N coordination. The FeN_5_ SAzyme had much higher POD-like activity than that of the FeN_4_ SAzyme, indicating that the introduction of axial N significantly enhanced the activity of SAzyme.

The doping of specific elements during the pyrolysis can effectively change the coordination environment of active metal centers in SAzymes and, thus, greatly influence their activities. For example, Jiao et al. [52] used FeCl_2_ as the Fe source, dicyandiamide as the N source, and boric acid as the B source to prepare B-doped Fe SAzyme (denoted as FeBNC) through pyrolysis. The incorporation of B triggered charge transfer to change the coordination environment of Fe, and thus greatly enhanced the POD-like activity of FeBNC, which provided a new idea for regulating the activity of SAzyme. Feng et al. [53] also prepared a B-doped Zn SAzyme (denoted as ZnBNC) by a similar method. The incorporation of B enhanced the N and O content, water dispersibility, and POD-like activity of ZnBNC, and it was also found that the incorporation of B could tune the catalytic activity by increasing defects.

To sum up, the strategies of changing the temperature and atmosphere of pyrolysis introducing axial coordination or doping during pyrolysis are essentially changing the coordination environment of the active center. Therefore, as long as it is a strategy that can change the coordination environment of the active centers of SAzymes, their activity can be regulated in principle, which lays the foundation for the development of more SAzymes activity regulation strategies.

Pyrolysis has been proved as an effective way to fabricate SAzymes, and their activity also can be facially regulated by tuning the temperature and atmosphere, introducing axial coordination as well as doping of specific elements. Thus, this strategy is widely used for the preparation of SAzymes for diverse applications. However, SAzymes prepared through this method are generally hydrophobic, so further surface modification is necessary for potential biomedical applications [20], and the particles may sinter together under the high temperature of pyrolysis. Thus, how to control the size of obtained SAzymes for a particular application is also a big challenge. Moreover, the pyrolysis process also consumes a lot of energy.

### 2.2. Defect Engineering

Baerlocher et al. [54] found that the presence of ordered Si vacancies significantly enhanced the catalytic activity of SSZ-74, and the formation of nanoscale ferroelectric domains in relaxor ferroelectrics was also associated with some form of structural disorder in the material induced by defects [55]. Therefore, controlling the generation of defects in materials may maximize the beneficial defects to improve their properties, that is, “defect engineering”. Wan et al. [56] synthesized a single-atom gold catalyst based on TiO_2_ nanosheets (denoted as Au–SA/Def–TiO_2_) (Figure 3A), which had abundant oxygen vacancy defects on its surface detected by electron paramagnetic resonance (EPR) measurement (Figure 3B). Au was effectively stabilized by the formation of Ti–Au–Ti, and the complete conversion temperature of Au–SA/Def–TiO_2_ catalyzed CO oxidation was lower than that of single-atom gold catalyst synthesized using perfect TiO_2_ without defects, indicating the higher catalytic activity brought by defects.

As a porous crystal, MOFs have inherent defects and complex structure [57], so they also have potential for SAzyme preparations by defect engineering. However, the introduction of defects may inevitably lead to structural instability of MOFs. Jasmina et al. [58] discovered zirconium-based MOFs (Zr-MOFs) UiO66, UiO67, and UiO68 in 2008. Due to the high strength of carboxylate–Zr bonds and the high connectivity of metal clusters [59], Zr-MOFs have excellent thermal [60], solvent [61], and high-pressure stabilities compared with other MOFs [62]. Therefore, Zr-MOFs have become the preferred material for the preparation of SAzymes by defect engineering. There are two main types of defects in MOFs (Figure 3C): the missing-linker defects and the missing-cluster defects [63,64]. Defects are introduced in a variety of ways, which can be divided into two broad categories: “de novo” synthesis and post-synthetic treatment [65,66,67]. “De novo” synthesis directly synthesizes MOFs with defects by changing the reaction conditions. Among them, the most widely used is the addition of modulators [66], including water [29], HCl [68], Hac [69], trifluoroacetate (TFA) [70], etc. The coordination ability of these modulators to clusters is much greater than that of organic ligands, so they will compete with organic ligands for coordination, resulting in defects. The content of defects in MOFs can be further tuned by changing the amount or type of modulators [68,71]. Post-synthesis treatments include post-synthesis exchange (PSE), the use of etchants, and so on. PSE, also known as solvent-assisted exchange, refers to the exchange of metal ions [72] or linkers [73] in MOFs. The etching method uses some acids, bases, and salts as etchants to introduce defects and even mesoporous or macroporous structures into MOFs, which can significantly adjust the performance of MOFs [74,75].

In the defects engineering strategy, defects are first introduced in MOFs by “de novo” synthesis or post-synthesis treatment, and then active ions or atoms are embedded in the defects. The distance between the metal nodes increases the distance between the defects; thus, the embedded ions or atoms are not easy to aggregate, ensuring the generation of atomically distributed active centers. Li et al. [68] used HCl and HAc as modulators to prepare defective NH_2_–UiO66 NPs (denoted as HCl–NH_2_–UiO66 NPs and Ac–NH_2_–UiO66 NPs, respectively), and then embedded Fe^3+^ in the defects to prepare the SAzymes with atomically dispersed Fe as the active center (denoted as Fe–HCl–NH_2_–UiO66 NPs and Fe–Ac–NH_2_–UiO66 NPs, respectively) (Figure 3D). The thermogravimetric curve (TGA) (Figure 3E) indicated that there were missing-linker defects in both Fe–HCl–NH_2_–UiO66 NPs and Fe–Ac–NH_2_–UiO66 NPs and more defects in Fe–HCl–NH_2_–UiO66 NPs. Fe–HCl–NH_2_–UiO66 NPs showed higher POD-like activity than Fe–Ac–NH_2_–UiO66 NPs and could be used for monitoring of trace H_2_O_2_ in cancer cells.

**Figure 3 molecules-27-05426-f003:**
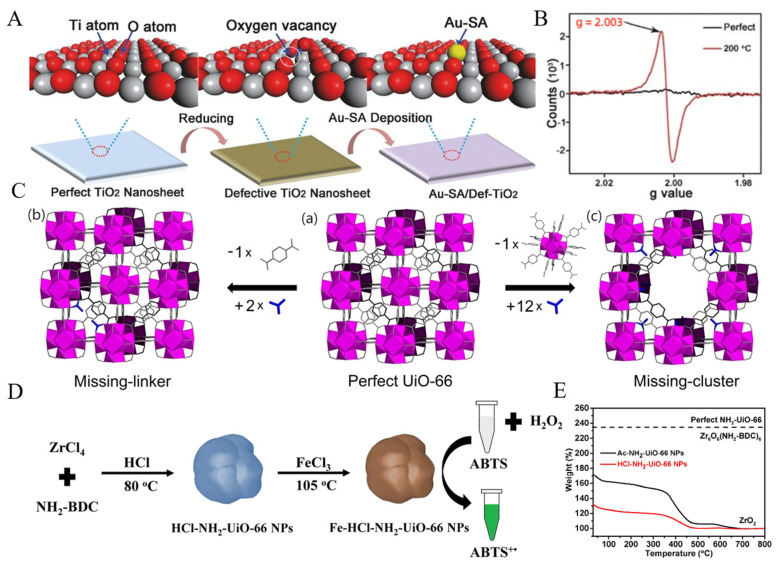
Defect engineering to prepare SAzymes. (**A**) Preparation of Au-SA/Def-TiO_2_ [56]. Copyright 2018, Wiley-VCH. (**B**) EPR of Per-TiO_2_ and Def-TiO_2_ [56]. Copyright 2018, Wiley-VCH. (**C**) Two kinds of defect formation in MOFs. (a) Perfect UiO-66. (b) Replacement of one linker with two monocarboxylic groups (in blue), generating one missing-linker defect per unit cell. (c) Replacement of one cluster with twelve monocarboxylic groups, generating one missing-cluster defect per unit cell. [59]. Copyright 2017, Elsevier. (**D**) Preparation of Fe–HCl-NH_2_-UiO66 NPs [68]. Copyright 2021, Elsevier. (**E**) TGA of Fe-HCl–NH_2_-UiO66 NPs and Fe-Ac-NH_2_-UiO66 NPs [68]. Copyright 2021, Elsevier.

The amount of modulators used can affect the content of defects and thus the properties of the catalyst. Ma et al. [76] prepared a series of defective NH_2_–UiO66 (denoted as UiO66–NH_2_–X, where X represented the molar ratio of HAc to linker) by adding different amounts of HAc during the synthesis. The content of defects increased with the increase amount of HAc. In the presence of the cocatalyst Pt (denoted as Pt@UiO66–NH_2_–X), the photocatalytic H_2_ production first increased and then decreased with the increase of structural defects, and Pt@UiO66–NH_2_–X exhibited the best catalytic activity and high stability when X was 100. Although Pt@UiO66–NH_2_–X is not an SAzyme, it shows that the content of defects can be changed by adjusting the amount of the modulator, thereby adjusting the activity of the catalyst, which contributes to a new and important idea for the activity regulation of SAzymes prepared by defect engineering.

Compared with the pyrolysis method, the defect engineering method can be conducted in a wet chemistry way; thus, the obtained SAzymes can be easily dispersed in aqueous solution and their original size can be largely preserved. This feature is quite important for the biomedical applications of SAzymes. However, the strategies to tune the activity of SAzymes fabricated through the defect engineering method are still greatly needed; now, SAzymes prepared by this way in the literature are mainly concentrated in industrial applications [56,77,78], and there are few reports on their biomedical applications. Nonetheless, it provides a new approach of low cost and low energy consumption to prepare SAzymes.

## 3. Characterizations of SAzymes

With the deepening of SAzymes research and the increasingly mature characterization techniques, more and more characterization techniques for SAzymes have emerged, including integrated electron microscopy, X-ray spectroscopy, infrared spectroscopy, nuclear magnetic resonance spectroscopy, etc. [33,41]. Herein, we focus on three characterization techniques that are most widely used in the field of SAzymes: high-angle annular dark-field scanning transmission electron microscopy (HAADF-STEM), XANES, and EXAFS.

If a characterization method can be used to directly see the uniformly distributed atomic-level active centers on the surface of the carrier, it will greatly promote the research of SAzymes. The electrons emitted inside the HAADF-STEM are partially scattered beyond the angle of convergence, and these high-energy electrons can be collected to image the isolated metal atoms [79]. Both XANES and EXAFS belong to X-ray absorption fine structure (XAFS) spectroscopy. XAFS establishes the relationship between X-ray absorption coefficient μ(E) and incident X-ray photon energy. The electrons of element atoms are liberated from lower-energy bound states, resulting in an increase in μ(E), and these energies are called the X-ray absorption edge of the element [80]. XANES is characterized by a signal of 30–50 eV above the X-ray absorption edge of a certain element atom, which reflects the valence of the atom and other information. EXAFS is characterized by the signals of 30–50 eV and 1000 eV above the absorption edges, providing information on the bonding structure around atoms. Structural information, the average atomic coordination number, interatomic distance, and other information can be obtained through Fourier transform-EXAFS (FT-EXAFS) and its fitting image, and wavelet transform-EXAFS (WT-EXAFS) can provide more structural information [27,81]. Examples of SAzymes characterization are shown in Table 2, and we choose two typical ones for detailed introduction.

For example, Su et al. [83] used Pluronic F127 as template, (NH_4_)_2_Fe(SO_4_)_2_ as Fe source, and dopamine as nitrogen source and carbon source to prepare N-doped mesoporous carbon nanospheres with atomically dispersed Fe as active center (denoted as SAFe–NMCNs) through pyrolysis. Atomically dispersed Fe (marked by red circles) could be directly observed in SAFe–NMCNs under the HAADF-STEM (Figure 4A). The coordination environment of Fe in SAFe–NMCNs was characterized by XANES and EXAFS. XANES (Figure 4B) results showed that the K-edge absorption of Fe in SAFe–NMCNs was between that of Fe foil and Fe_2_O_3_, indicating that it was positively charged and had a valence state between 0 and +3. EXAFS (Figure 4C) results indicated that Fe–N scattering paths (1.55 Å) and no Fe–Fe scattering paths (2.2 Å) existed in SAFe–NMCNs. At the same time, the results of WT-EXAFS (Figure 4D) also showed that there was only Fe–N but no Fe–Fe in SAFe–NMCNs, which further indicated that Fe was atomically dispersed. After fitting, the coordination number of Fe is 4.23, indicating that the Fe–N_4_ structure was formed, which meant that Fe in SAFe–NMCNs was coordinated by four N atoms.

Li et al. [68] prepared Fe–HCl–NH_2_–UiO66 NPs using defect engineering. No formation of Fe nanoparticles was observed in the HAADF-STEM image, and the element mapping image also confirmed the uniform distribution of Fe in Fe–HCl–NH_2_–UiO66 NPs. Moreover, the X-ray photoelectron spectroscopy of Fe 2p showed that Fe(III) characteristic peaks existed in Fe–HCl–NH_2_–UiO66 NPs, indicating that Fe had +3 valence. Then, using XAFS for further analysis, the results showed that Fe–HCl–NH_2_–UiO66 NPs had similar absorption to Fe_2_O_3_, which further indicated that Fe had a +3 valence. The coordination environment of Fe was analyzed by FT-EXAFS and the fitting results, which showed coexistence of Fe–O and Fe–Cl and no existence of Fe–Fe, indicating that Fe in Fe–HCl–NH_2_–UiO66 NPs was connected to Zr_6_ clusters through Fe–O–Zr.

## 4. Surface Modifications of SAzymes

The catalytic process of nanozymes is different to that of natural enzymes. The catalyzed process of a natural enzyme is (a) substrate binding, (b) catalytic reaction, (c) product release, while that of a nanozyme is similar to heterogeneous catalytic reaction: (a) substrate adsorption, (b) surface reaction, (c) product dissociation and surface active site regeneration [23,84]. Thus, the catalytic process of nanozymes is closely related to their surface [85]. At the same time, since nanozymes are mostly inorganic nanoparticles or carbon materials containing metal elements, their colloidal stability, biocompatibility, and targeting properties need to be improved for biomedical applications [20]. For this purpose, the best choice is the surface modification [30].

For example, polyethylene glycol (PEG) is often used to improve hydrophilicity and biocompatibility of materials. Huo et al. [86] immobilized Fe^III^ acetylacetone in ZIF-8 by hydrothermal method, followed by pyrolysis in Ar to obtain Fe SAzymes with N-doping carbon material as the carrier (denoted as SAF NCs) (Figure 5A) for tumor treatment. To enhance their hydrophilicity and biocompatibility, DSPE–PEG–NH_2_ was modified on the surface of SAF NCs through hydrophobic–hydrophobic interactions to obtain PEG-modified Fe SAzymes (denoted as PSAF NCs) (Figure 5B). After modification, the distribution of particle size was reduced, the surface potential was more negative, and the dispersion in normal saline was better (Figure 5C). Besides PEG, polyvinyl pyrrolidone is also often used as a surface modifier to improve the dispersity and biocompatibility of SAzymes [87].

In addition to enhancing the dispersity and biocompatibility of SAzymes, surface modification also helps to improve targeting. Gong et al. [88] synthesized carbon dots with citric acid and polyene polyamines, which were subsequently loaded with HAuCl_4_ and reduced with NaBH_4_ to obtain SAzyme with an active center of Au (denoted as CAT-g). Triphenyl phosphorus (TPP) and cinnamaldehyde (CA) were then modified on its surface, and the modified CAT-g was denoted as MitoCAT-g. TPP could target mitochondria, Au depleted glutathione (GSH) in mitochondria, and CA produced ROS. MitoCAT-g destroyed the redox balance of tumor cells, amplified the effect of ROS to destroy mitochondria, and led to apoptosis, so as to achieve the purpose of tumor treatment. The modification of TPP resulted in the distribution of more particles in mitochondria; thus, they could kill cancer cells more effectively.

The method of cell membrane modification is also widely used in the field of SAzyme. Liu et al. [89] prepared Fe SAzymes (denoted as SAF NPs) by pyrolyzing ZIF-8 loaded with Fe species (Figure 6A). After loaded doxorubicin (DOX) within their porous structure, SAF NPs were modified by human non-small-cell lung cancer cell membrane (A549 CM), denoted as SAF NPs@DOX@CM (Figure 6B). There were specific proteins on the surface of cancer cell membrane (CM), and the influence of homology enabled CM-modified NPs to escape the phagocytosis and immune rejection of macrophages, thereby greatly prolonging their blood circulation time and enhancing tumor targeting. After incubated with human normal hepatocytes and A549 cells for 24 h, respectively, both SAF NPs@CM and SAF NPs@DOX@CM caused extensive apoptosis of A549 cells and the DOX-loaded group caused more (Figure 6C), which was related to the targeting ability of NPs after CM modification and the difference in pH of tumor cells from normal cells. The in vivo experiment also proved that CM modification endowed NPs with the ability to target tumors (Figure 6D).

Similarly, Qi et al. [90] used platelet membrane (PM) to encapsulate a pyrolyzed mesoporous Fe SAzyme (denoted as PMS). The more negative potential of PMS indicated the successful modification of PM. PMS showed good POD-like activity and photothermal effect, which was expected to realize the combination of chemodynamic therapy and photothermal therapy. More importantly, there is an important protein in PM: P-selectin, which can target tumor cells, thus PMS was selectively endocytosed by 4T1 cells. Meanwhile, PMS was not phagocytosed by Raw264.7 cells, indicating that the homology of PM enabled Fe SAzyme to escape from macrophage phagocytosis.

At present, the surface modification strategies for SAzymes are mostly learned from those already built in the fields of nanomedicine and show great success. However, one should pay special attention to the influence of surface modification on the activity of nanozymes, because the following applications greatly depend on their activity. For example, Sanjay et al. [91] found that CAT-like activity of cerium oxide nanoparticles (CeNPs) increased after they were modified with PO_4_^3−^, but when the concentration of PO_4_^3−^ exceeded 100 μM, the SOD-like activity of CeNPs would be significantly inhibited. It was speculated that the inhibition of SOD-like activity might be caused by the reaction of PO_4_^3−^ with CeNPs to produce products similar to cerium phosphate. Therefore, the selection of modifiers during surface modification needs careful consideration. More recently, Wang et al. [92] demonstrated that membrane cholesterol depletion could enhance enzymatic activity of cell-membrane-coated MOF NPs. The mechanism behind this phenomenon is that the reducing cholesterol level effectively enhances membrane permeability, thus the substrates are more accessible to the encapsulated enzymes. These findings can provide facile and practical ways for the modulation of the activity of SAzymes through surface modification.

## 5. Applications of Single-Atom Nanozymes in Biomedicine

Due to its high atom utilization, unsaturated coordination of active centers, and geometric structures similar to those of nature enzymes, SAzymes are widely used in biosensing [40,42,43,82,93,94], cancer treatment [48,51,83,95,96], and so on. In previous review articles, these two types of applications have been comprehensively discussed; thus, this review will not repeat them. Herein, we mainly focus on the applications of SAzymes in ROS scavenging and antibacterial, and the specific mechanism is shown in Figure 7. For ROS scavenging, normally the CAT and SOD-like SAzymes are used, because they can eliminate the excess ROS and alleviate oxidative stress. However, for the antibacterial applications, ROS are produced by SAzymes to damage the membranes or biomacromolecules of bacteria, resulting in the death of them, finally.

### 5.1. SAzymes for ROS Scavenging

O_2_ participates in thousands of reactions in the human body. Under the catalysis of enzymes in mitochondria, it is reduced to water by transferring four electrons and generates adenosine triphosphate for energy [97], but sometimes a single-electron or double-electron transfer reaction occurs to generate ROS, mainly including •O_2_^−^, H_2_O_2_, and •OH [98]. ROS is a double-edged sword. Low doses of ROS are essential for the regulation of life, such as cell division [99], signal transmission [100], and so on. However, when the level of ROS exceeds the normal level, it will cause oxidative damage to cells, resulting in hair loss [101], inflammation [46], stroke [14], Parkinson’s [12], and other diseases. The level of ROS in the human body is always at a relatively stable level, which depends on the interaction of four enzymes: OXD, POD, CAT, and SOD [102]. OXD and POD can increase ROS, while CAT and SOD can reduce ROS. Therefore, SAzymes with CAT-like and SOD-like activities are expected to scavenge excess ROS and alleviate cellular oxidative damage.

Ma et al. [103] used ZIF-8 to encapsulate FePc followed by pyrolysis to obtain SAzymes with atomically dispersed Fe as the active center (denoted as Fe–SAs/NC) and reported their CAT-like and SOD-like activities. Fe–SAs/NC was modified with DSPE–PEG_2000_ to improve their biocompatibility. The cell experiments proved that Fe–SAs/NC could scavenge ROS and alleviate cellular oxidative damage. Similarly, Lu et al. [104] utilized the pyrolysis of Fe-TPP ⊂ rho-ZIF (Fe-TPP = tetraphenylporphyrin iron; rho-ZIF = zeolitic imidazolate skeleton with rho topology) to obtain atomically dispersed Fe SAzymes (denoted as Fe–N/C SACs). Fe–N/C SACs had multi-enzyme-like activities, including OXD-like, POD-like, CAT-like, and GPx-like activities, which could alleviate cellular oxidative damage.

Yan et al. [105] used CeO_2_ as the carrier to load atomically dispersed Pt and obtained Pt@CeO_2_ SAzyme. Then, they compounded the SAzyme with polyacrylonitrile fiber and medical polyethylene tape to prepare a Pt@CeO_2_ SAzyme-based bandage (Figure 8A) to scavenge excessive ROS and reactive nitrogen species around the traumatic brain injury (TBI) wound in order to alleviate neuroinflammation. Pt@CeO_2_ SAzyme had POD-like, CAT-like, SOD-like, and GPx-like activities; thus, they had excellent ability of scavenging ROS inside cells (Figure 8B). At the same time, Pt@CeO_2_ was less cytotoxic and could alleviate the oxidative damage of cells caused by H_2_O_2_ and lipopolysaccharide (LPS) (Figure 8C). Animal experiments showed that the TBI wound healing effect of the Pt@CeO_2_ treatment group was significantly better than that of the other groups, indicating the correctness and feasibility of the bandage treatment principle. Similarly, the Co/PMCS also showed SOD-like, CAT-like, and GPx-like activities [46]. It could effectively reduce the content of ROS caused by various stimuli to alleviate oxidative damage of cells, and the animal experiments showed that whether LPS or *Escherichia coli* (*E. coli*) were used to induce sepsis in mice, Co/PMCS had a good therapeutic effect, and the levels of tumor necrosis factor-α and interleukin- 6 were significantly decreased.

In addition to the ability of SAzyme to simulate CAT, SOD, and GPx to alleviate cellular oxidative damage, Chen et al. [106] used electrochemical deposition to deposit Cu on g-C_3_N_4_ and obtained a kind of SAzyme in which 1 Cu atom was coordinated with four N atoms (denoted as Cu–SAs/CN). Cu–SAs/CN possessed ascorbate peroxidase (APX)-like activity, which can decompose H_2_O_2_ in the presence of AsA, thereby alleviating cellular oxidative damage. The Cu–SAs/CN had almost no cytotoxicity, and could alleviate H_2_O_2_-induced cell damage.

Although there are some reports on SAzymes that can scavenge ROS and alleviate cellular oxidative damage, they are limited to a few materials, so it is still necessary to expand material systems. At the same time, the intrinsic mechanisms of ROS scavenging for SAzymes are still relatively vague, and great efforts need to be paid in this direction.

### 5.2. SAzymes for Antibacterial

Pathogenic bacteria infection is a major threat to human health globally [107]. Over the past few decades, antibiotics, such as penicillin, have been widely used clinically as antibacterial agents [108]. In recent years, nanozymes with antibacterial activities are regarded as a novel bactericide, due to their negligible biotoxicities, no drug resistance, and broad-spectrum antibacterial performance [109]. SAzymes with highly efficient catalytic activities also play an important role in antibacterial treatments [22].

Xia et al. [110] successfully prepared atomic-level Ag-loaded MnO_2_ porous hollow microspheres (Ag/MnO_2_ PHMs) through the redox precipitation process, which exhibited superior photothermocatalytic inactivation of *E. coli* under solar light irradiation. On the one hand, atomic Ag with high conductivity increased the level of Mn^3+^ and oxygen vacancies to excite MnO_2_, thus promoting the introduction of reactive species for photocatalysis. On the other hand, atomic Ag enhanced the photothermal conversion and lattice oxygen reducibility, thus promoting thermocatalysis.

Most SAzymes utilize their POD-like activities to generate toxic •OH, therefore achieving efficient sterilization. For instance, Xu et al. [47] fabricated the PMCS via an m-SiO_2_-protected pyrolysis approach (Figure 9A). Owing to the high POD-like activity, PMCS showed outstanding antibacterial performance against *Pseudomonas aeruginosa* (Figure 9B). The wound caused by *Pseudomonas aeruginosa* healed well (Figure 9C, Figure 9D), indicating that PMCS had good antibacterial properties.

As reported by Huo et al. [111], nanocatalysts of single iron atoms anchored in nitrogen-doped amorphous carbon (SAF NCs) were synthesized via encapsulated-pyrolysis strategy. SAF NCs performed excellent POD-like activities in the presence of H_2_O_2_, producing toxic •OH to achieve high-efficiency sterilization effect against *E. coli* and *Staphylococcus aureus* (*S. aureus*). With the assistance of near-infrared light, antibacterial properties were further improved due to the intrinsic photothermal property of SAF NCs. Noticeably, the antibacterial mechanisms of crucial CM destruction induced by SAF NCs were also revealed. Wang et al. [112] prepared a novel Cu single-atom sites/N-doped porous carbon (Cu SASs/NPC) with POD-like activity by pyrolysis–etching–adsorption–pyrolysis strategy. The doping of Cu considerably enhanced POD-like activity and accelerated GSH-depleting and photothermal properties. The synergistic effect made Cu SASs/NPC exhibit excellent antibacterial performance against *E. coli* and methicillin-resistant *S. aureus* (MRSA). Remarkably, by reversing the thermal sintering process, Chen et al. [113] proposed the direct transformation of Pt NPs into Pt single atoms to gain Pt SAzyme (PtTS–SAzyme). It showed excellent POD-like catalytic activity, which was attributed to the unique structure with Pt_1_–N_3_PS active moiety. Compared with Pt NPs, PtTS–SAzyme exhibited highly efficient antibacterial performance and broad-spectrum antibacterial properties. In addition to POD-like activity, SAzymes with OXD-like activity were also used in antibacterial applications. Huang et al. [50] developed SAzymes with carbon nanoframe–confined FeN_5_ active centers (FeN_5_ SA/CNF), which showed effective antibacterial effects against *E. coli* and *S. aureus*.

Besides photothermal effect, Yu et al. [114] innovatively fabricated a red blood cell membrane modified Au nanorod-actuated single-atom-doped porphyrin metal−organic framework (denoted as RBC–HNTM–Pt@Au) with an excellent sonocatalytic activity. Under ultrasound, not only was the antibacterial activity of RBC–HNTM–Pt@Au greatly enhanced, but SAzymes were also directionally propelled, which played a significant part in the treatment of MRSA-infected osteomyelitis.

In summary, the oxidation effect of SAzymes, especially POD-like activity, may play a major role in antibacterial applications. The ROS, such as •OH, are extremely toxic to the bacteria cells usually by disrupting the integrity of the cell membrane. However, this effect is nonselective; thus, normal cells may also be damaged. To avoid such kind of side effect, the design of smart SAzymes whose activity only can be activated on the infection sites is significantly important. The whole applications mentioned in this section are summarized in Table 3.

## 6. Summary and Outlook

SAzymes with excellent performance have been widely used in the field of biomedicine. Compared with ordinary nanozymes, they have defined active sites and coordination environments, which are important for understanding the relationship between their structures and activities. It is also beneficial for the research of the mechanism and the catalytic process in order to better mimic the natural enzyme. At the same time, atom utilization is greatly improved due to the atomic-level dispersion of active sites. In this review, we summarized the preparation and characterization of SAzymes, discussed their surface modification, and finally focused on their applications of ROS scavenging and antibacterial.

Although significant advances have been achieved in this area, challenges are still remaining. First of all, new fabrication strategies should always be pursued to obtain SAzymes with new structures and activities, especially those that have ROS scavenging activity. The preparation of SAzymes needs theoretical guidance. We can further combine theoretical calculation, big data, artificial intelligence, and other technologies to guide the design and preparation of SAzymes. Furthermore, the strategies which can combine the advantages of finely regulated structures and activity with large-scale and green production processes are highly welcomed. Moreover, the surface modification is necessary for SAzymes in biomedical applications; however, the basic knowledge of how different surface modifications would influence their catalytic activities is largely unknown. Finally, for the biomedical applications of SAzymes, their long-term in vivo biodegradability and toxicity should be carefully investigated, which is also crucial for their real applications. We believe these challenges can be addressed in the future through the collaborations of researchers from different disciplines, such as chemistry, material science, computer science, and biology, as well as medicine.

## Figures and Tables

**Figure 2 molecules-27-05426-f002:**
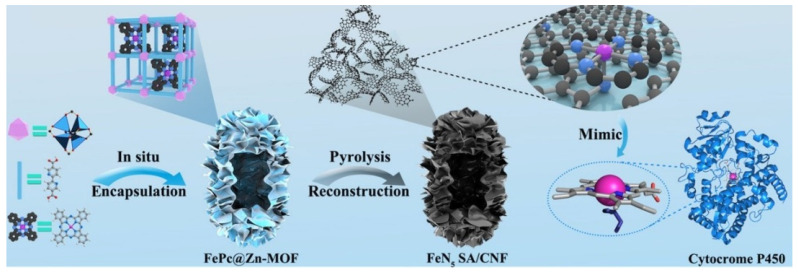
Schematic formation process of carbon nanoframe–confined atomically dispersed Fe sites with axial five-N coordination for mimicking the active center of cytocrome P450 [50]. Copyright 2019, American Association for the Advancement of Science.

**Figure 4 molecules-27-05426-f004:**
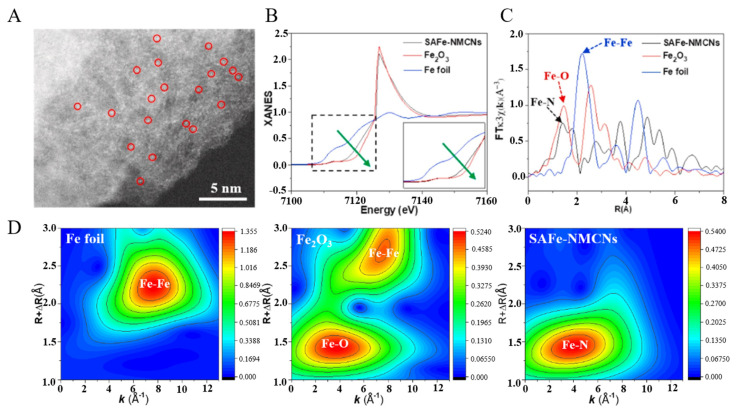
Structure characterization of SAFe-NMCNs (**A**) HAADF-STEM. (**B**) XANES. (**C**) FT-EXAFS. (**D**) WT-EXAFS [83]. Copyright 2022, Elsevier.

**Figure 5 molecules-27-05426-f005:**
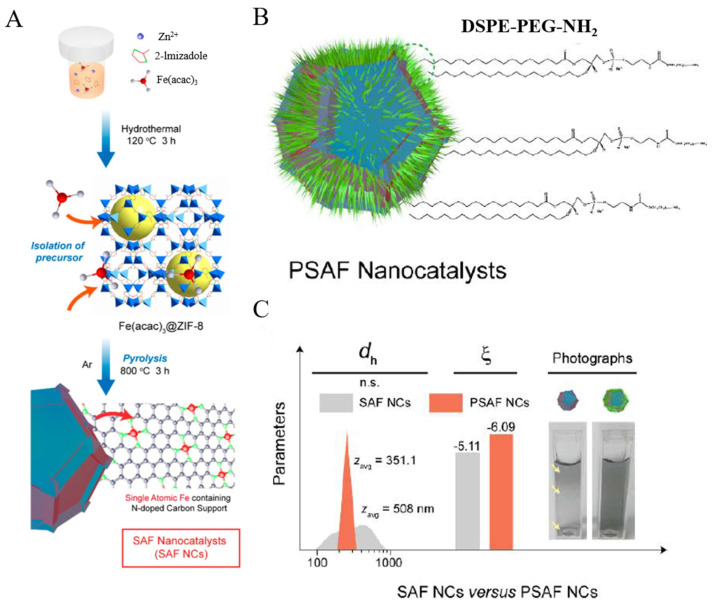
(**A**) Preparation of SAF NCs. (**B**) Structure of PSAF NCs. (**C**) Particle size distribution, zeta potential, and dispersibility in normal saline [86]. Copyright 2019, American Chemical Society.

**Figure 6 molecules-27-05426-f006:**
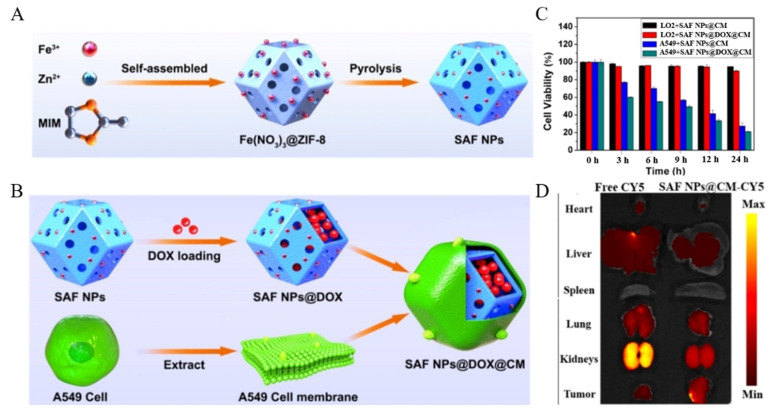
Preparation of SAF NPs (**A**) and SAF NPs@DOX@CM (**B**). (**C**) Viability of cells treated with SAF NPs@CM and SAF NPs@DOX@CM. (**D**) Fluorescence intensities of major organs and tumors after 24 h intravenous injection of free CY5 and SAF NPs@CM-CY5 [89]. Copyright 2021, Wiley-VCH.

**Figure 7 molecules-27-05426-f007:**
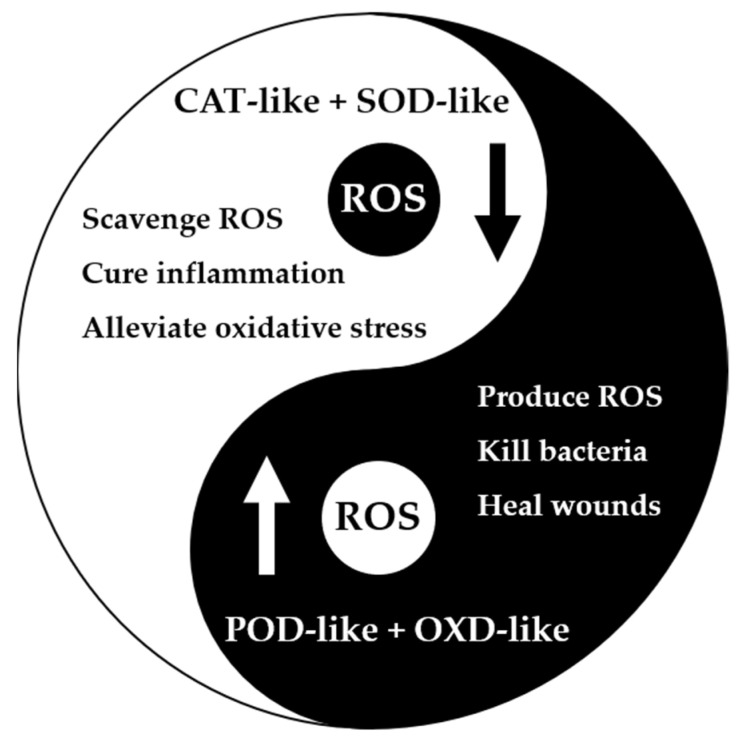
Mechanism of SAzymes for scavenging ROS and antibacterial.

**Figure 8 molecules-27-05426-f008:**
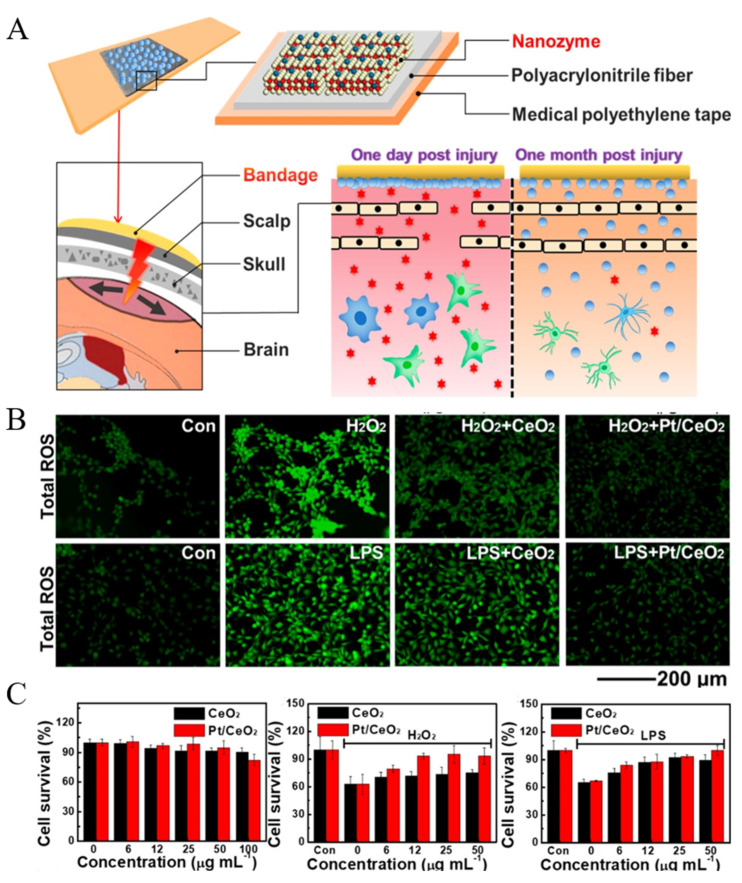
(**A**) Design principle of Pt@CeO_2_ bandage. (**B**) ROS scavenging abilities of Pt/CeO_2_ inside cells. (**C**) Alleviating cellular oxidative damage abilities of Pt/CeO_2_ [105]. Copyright 2019, American Chemical Society.

**Figure 9 molecules-27-05426-f009:**
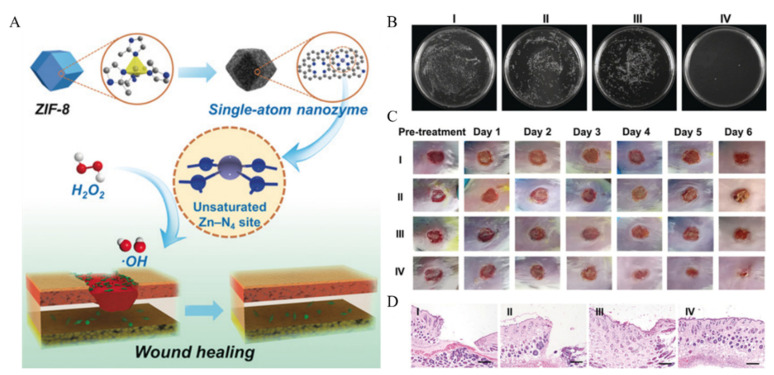
(**A**) Preparation of PMCS. (**B**) Antibacterial properties of PMSC. (**C**) Abilities of PMCS to promote wound healing. (**D**) Histologic analysis of the wounds (I: NaAc buffer, II: NaAc buffer + H_2_O_2,_ III: PMCS, and IV: PMCS + H_2_O_2_) [47]. Copyright 2019, Wiley-VCH.

**Table 1 molecules-27-05426-t001:** Advantages and disadvantages of preparation methods of SAzymes.

Methods	Advantages	Disadvantages	Ref.
Pyrolysis	Widely used, high active centers loading, well-defined structure, large-scale manufacture potential.	High energy consuming, uncontrollable particle size, poor biocompatibility.	[20,30]
Defect engineering	Better biocompatibility, controllable particle size, low cost, low energy consuming.	Less application system, more activity modulation methods need to be developed.	[27]
Atomic layer deposition	Precise control of active center deposition, convenient to study the relationship between catalyst structure and performance.	Difficult to achieve mass production, high cost.	[34]
Photochemical reduction	Easy to operate, no professional equipment required.	Relatively low active center loading, difficult to achieve mass production.	[35,36]

**Table 2 molecules-27-05426-t002:** Summary of characterizations of SAzymes.

SAzymes	Characterization	Results	Ref.
Fe–N–C SACs	HAADF-STEM	Atomically dispersed Fe	[82]
	XANES	Valence state of Fe was between 0 and +3	
	EXAFS	Only Fe–N existed	
Co/PMCS	HAADF-STEM	Atomically dispersed Co	[46]
	XANES	Co was positive charged	
	EXAFS	CoN_4_ existed	
FeN_5_ SA/CNF	HAADF-STEM	Atomically dispersed Fe	[50]
	XANES	Contained FeN_4_ structure	
	EXAFS	FeN_5_ existed	
ZnBNC	HAADF-STEM	Atomically dispersed Zn	[53]
	XANES	Valence state of Zn was between 0 and +2	
	EXAFS	ZnN_4_ existed	
Au–SA/Def–TiO_2_	HAADF-STEM	Atomically dispersed Au	[56]
	XANES	Valence state of Au was +3	
	EXAFS	Au–O and Au–Ti existed	
Fe–HCl–NH_2_–UiO66 NPs	HAADF-STEM	Atomically dispersed Fe	[68]
	XANES	Valence state of Fe was between +3	
	EXAFS	Fe–O–Zr existed	
SAFe–NMCNs	HAADF-STEM	Atomically dispersed Fe	[83]
	XANES	Valence state of Fe was between 0 and +3	
	EXAFS	FeN_4_ existed	

**Table 3 molecules-27-05426-t003:** Summary of applications of SAzymes in ROS scavenging and antibacterial.

Applications	SAzymes	Enzyme-Like Activities	Ref.
ROS scavenging	Fe–SAs/NC	CAT, SOD	[103]
	Fe–N/C SACs	OXD, POD, CAT, GPx	[104]
	Pt@CeO_2_	POD, CAT, SOD	[105]
	Co/PMCS	SOD, CAT, GPx	[46]
	Cu–SAs/CN	APX	[106]
Antibacterial	Ag/MnO_2_ PHMS	/	[110]
	PMCS	POD	[47]
	SAF NCs	POD	[111]
	Cu SASs/NPC	POD	[112]
	PtTS–SAzyme	POD	[113]
	FeN_5_ SA/CNF	OXD	[50]
	RBC–HNTM–Pt@Au	/	[114]

## Data Availability

Not applicable.

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
