# Peer review of "Single-Atom Nanozymes: Fabrication, Characterization, Surface Modification and Applications of ROS Scavenging and Antibacterial"

_molecules, 2022, doi:10.3390/molecules27175426_

Round 1

Reviewer 1 Report

The MS entitled “Single-atom Nanozymes: Fabrication, Characterization, Surface  Modification and Applications of ROS Scavenging and Anti-bactial” is interesting, Single atom nanoenzyme is gaining interest within the scientific community, thus review article on this topic gaining popularity. Though the MS requires improvements. Some of the observations are as follows:

1.       Spelling mistake in Title is not appreciable.

2.       Abstract should be re-written in a more technical way so that authors could trace the paper easily.

3.       MOF strategy has been covered, but it should be covered as a separate sub-heading.

4.       Atom trapping strategy, one-pot high-temperature calcination method, and post-modification strategy should be covered in detail and with proper diagrams.

5.       The characterization portion needs more literature.

6.        English language editing is required.

7.       The application part is poorly constructed, the detailed mechanism along with the necessary figures should be included.

8.       Tables for the synthesis method, advantages and disadvantages of each of the methods should be included in the tables.

9.       A detailed table for characterization and another table for application is required.

Author Response

Point 1: Spelling mistake in Title is not appreciable.

Response 1: We apologize for this mistake! The spelling mistake in the title has been corrected from ’antibactial’ to ‘antibacterial’ (Page 1, the title).

Point 2: Abstract should be re-written in a more technical way so that authors could trace the paper easily.

Response 2: Abstract has been rewritten in a more technical way in order to make it convenient for readers to trace. Please see the abstract for detials. (Page 1 ,the abstract)

Point 3: MOF strategy has been covered, but it should be covered as a separate sub-heading.

Response 3: Thank you for your suggestions! In this review, we classify the preparation methods of SAzymes according to the fabrication process, not to the starting materials. Thus both pyrolysis and defect engineering methods described in the section of ‘preparation of SAzymes’ include MOF strategies. Therefore, if MOF strategies are covered as a separate sub-heading alone, it will make the classification confusing. So the MOF strategy is not covered as a separate sub-heading, but is described in both pyrolysis and defect engineering methods.

Point 4: Atom trapping strategy, one-pot high-temperature calcination method, and post-modification strategy should be covered in detail and with proper diagrams.

Response 4: Thank you for your suggestions! As mentioned in point 3, the preparation methods can be classified according to different standards. In this review, we classify the methods according to the fabrication process, focusing on pyrolysis and defect engineering methods. Actually, the atomic trapping strategies are used in defect engineering and atomic layer deposition. One-pot high temperature calcination method is what we refer to as pyrolysis. The post-modification strategy includes the insertion of atoms and the exchange of ligands in defect engineering. So, the three strategies you mentioned all have their counterparts in our article, just expressed in different ways. At the same time, some of the methods have been extensively introduced in details in previous excellent reviews. In order to avoid excessive repetition and to propose new contents, we focus on pyrolysis and defect engineering methods here.

Point 5: The characterization portion needs more literature.

Response 5: We have added 5 additional literatures in the characterization portion (ref. 42, 50, 53, 56, and 82). (Page 19 and Page 20, text in red)

Point 6: English language editing is required.

Response 6: The English language has been careffuly editted and polished. Some major changes have been maked in red but those deleted are not shown. Please see the text for details.

Point 7: The application part is poorly constructed, the detailed mechanism along with the necessary figures should be included.

Response 7: We have added a figure named ‘Figure 7. Mechanism of SAzymes for scavenging ROS and antibacterial.’ to describe the detailed mechanism. Please see the Figure7 and the text for details.(Page 12, Figure 7 and text in red)

Point 8: Tables for the synthesis method, advantages and disadvantages of each of the methods should be included in the tables.

Response 8: We have added a table named ‘Table 1. Advantages and disadvantages of preparation methods of SAzymes.’ to summarize the advantages and disadvantages of each of the methods. (Page 2, Table 1)

Point 9: A detailed table for characterization and another table for application is required.

Response 9: We have added two tables named ‘Table 2. Summary of characterizations of SAzymes.’ (Page 8, Table 2) And ‘Table 3. Summary of applications of SAzymes in ROS scavenging and antibacterial’(Page 15, Table 3) to summarize the characterizations and biomedical appications of SAzymes.

Reviewer 2 Report

Thanks a lot for your invitation to review this manuscript "Single-atom Nanozymes: Fabrication, Characterization, Surface  Modification and Applications of ROS Scavenging and Antibacterial" (molecules-1860743). This manuscript has not been able to thoroughly and comprehensively investigate Single-atom Nanozymes. Also, there is a lot of overlapping of the given information with previously published articles (https://dx.doi.org/10.1021/acs.analchem.0c04084?ref=pdf, DOI: 10.1126/sciadv.aav5490, and DOI: 10.1039/D0BM01447H) reducing the manuscript's innovation level. therefore I recommend rejection of the manuscript.

Author Response

Point 1: Thanks a lot for your invitation to review this manuscript "Single-atom Nanozymes: Fabrication, Characterization, Surface  Modification and Applications of ROS Scavenging and Antibacterial" (molecules-1860743). This manuscript has not been able to thoroughly and comprehensively investigate Single-atom Nanozymes. Also, there is a lot of overlapping of the given information with previously published articles (https://dx.doi.org/10.1021/acs.analchem.0c04084?ref=pdf, DOI: 10.1126/sciadv.aav5490, and DOI: 10.1039/D0BM01447H) reducing the manuscript's innovation level. therefore I recommend rejection of the manuscript.

Response 1: We have carefully read the two articles you mentioned. It is ture that our paper has some overlaps with them, because single-atom nanozymes (SAzymes) are still a relatively new research field, and the literatures that can be found are limited, so some classic and commonly used methods mentioned in previous review articles should also be mentioned in our article. However, more newer literatures are introduced and disscussed in our paper, thus the readers can get the new advances about these classic methods. Actually, in order to avoid more overlapping, in this paper we focus on the issues which are important for the biomedical applications of SAzymes but less discussed in previous review articles. The readers can find other useful informations from previous excellent review papers.

In the two articles you mentioned, the authors both classify SAzymes according to the materials, including the type of active centers and the type of carriers. At the same time, the preparation of SAzymes by defect engineering and the regulation of activity are not mentioned in either these two articles or many other review papers. Our paper focuses on these two parts, which can also provide new knowledges for readers. Finally, we focus on the surface modification of SAzymes used in the biomedical field, including improving targeting, biocompatibility, etc., and we discuss the effect of surface modification on the activity of SAzymes. These aspects are very important for the biomedical applications of SAzymes, but to our best knowledge they are not mentioned in previous review papers.

Reviewer 3 Report

I would like to thank the author for this vital topic, which aims to improve the level of disinfection & ROS scavenging.

1- The authors should prepare an abbreviation list.

2- For the preparation of single-atom nanozymes, the authors have been mainly focus on the widely used pyrolysis methods and the wet chemistry. The Preparation methods should be present in the abstract.

2- Keywords should be revised. Some of them are very general, For example “

 surface modification”.

3- Disadvantages of the single-atom nanozymes should be present in the manuscript.

4- The genus and species of bacteria should be written in italics form, for example; E.coli, Pseudomonas aeruginosa, Staphylococcus aureus

5- Recently, some scientific reports have been published on the use of Ni-Fe/Fe3O4 magnetite for disinfection of E. coli. The authors should mention it in the disinfection section.

Author Response

Point 1: The authors should prepare an abbreviation list.

Response 1: We have added an abbreviation list at the end of our article.(Page 16-17 Abbreviations)

Point 2: For the preparation of single-atom nanozymes, the authors have been mainly focus on the widely used pyrolysis methods and the wet chemistry. The Preparation methods should be present in the abstract.

Response 2: We have added the preparation methods in the abstract. (Page 1, the abstract)

Point 3: Keywords should be revised. Some of them are very general, For example ‘surface modification’.

Response 3: Thank you for your suggestions! In our article, surface modification includes three parts: hydrophilic polymer modification, targeted molecular modification and cell membrane modification, mainly to increase the biocompatibility, targeting, long-term circulation of SAzymes in the body. The term ‘surface modification’ can summarize all these three parts. If it is divided into more specific words such as PEG modification, targeting or cell membrane modification, it will exceed the number limitation of keywords, about five. So we think although ‘surface modification’ is general, it may be suitable to summarize the three parts mentioned in our paper.

Point 4: Disadvantages of the single-atom nanozymes should be present in the manuscript.

Response 4: The disadvantages of single-atom nanozymes are discussed in the last paragraph of the section of ‘summary and outlook’, and the challenges they face are the disadvantages, including the urgent development of new synthetic strategies and material systems, the lack of theoretical guidance, active centers confined to metal elements, poor long-term circulation stability in the body, and possible toxic and side effects. (Page 16, Last paragraph, text in red)

Point 5: The genus and species of bacteria should be written in italics form, for example; E.coli, Pseudomonas aeruginosa, Staphylococcus aureus

Response 5: The genus and species of bacteria have been corrected into italics form. (Page 13 Line 491, Page 14 Line 521/529/530, Page 15 Line 541/542/549/550/558, and text in red)

Point 6: Recently, some scientific reports have been published on the use of Ni-Fe/Fe3O4 magnetite for disinfection of E. coli. The authors should mention it in the disinfection section.

Response 6: Thank you for your suggestions! According to the keywords you provided, we retrieved this article titled 'Electrocatalytic disinfection of E. coli using Ni-Fe/Fe3O4 nanocomposite cathode : Effect of Fe3O4 nanoparticle, humic acid, and nitrate' ( DOI : 10.1016/j.seppur.2022.121140). After careful reading, we found that it used Ni-Fe/Fe3O4 as an electrode for electrocatalytic disinfection. Ni-Fe/Fe3O4 is not a single-atom nanozyme, thus this paper has little relevance to the theme of our article, so we think it is unnecessary to mention it in the section ‘SAzymes for antibacterial’.

Round 2

Reviewer 1 Report

Acceptable

Reviewer 2 Report

This manuscript after native English revision could be acceptable for publication.